# Numerical Simulation for Durability of a Viscoelastic Polymer Material Subjected to Variable Loadings Fatigue Based on Entropy Damage Criterion

**DOI:** 10.3390/polym16202857

**Published:** 2024-10-10

**Authors:** Yutong Li, M. J. Mohammad Fikry, Jun Koyanagi

**Affiliations:** 1Department of Materials Science and Technology, Tokyo University of Science, 6-3-1 Niijuku, Katsushika-ku, Tokyo 125-8585, Japan; liyutong@rs.tus.ac.jp; 2Department of Mechanical Engineering, The University of Akron, 244 Sumner St., Akron, OH 44325-3903, USA; mfikry@uakron.edu

**Keywords:** numerical simulation, viscoelastic polymer materials, fracture fatigue entropy (FFE), variable amplitude, fatigue life

## Abstract

This study aims to explore the impact of load history on the premature failure of the viscoelastic polymer matrix in carbon-fiber-reinforced plastics (CFRPs) using a method based on the concept of fracture fatigue entropy (FFE). A user-defined subroutine (UMAT) developed by the authors in previous studies was incorporated to apply the FFE damage criterion using ABAQUS software. Several variable-amplitude load modes, including frequent load amplitude changes and intermittent interruptions, were designed based on the conventional linear damage accumulation method (Palmgren–Miner rule), and the fatigue life under these loadings was obtained via numerical simulations. The results show that both frequent amplitude changes and even brief pauses in loading can accelerate damage accumulation, leading to premature failure of the polymer matrix. In these scenarios, the fatigue life ranged from 33.6% to 91.9% of the predictions made using the Palmgren–Miner rule, which shows significant variation and highlights cases in which the predicted fatigue life falls far short of expectations. This study offers a more practical and reliable approach for predicting fatigue life under complex loading conditions. Since the accuracy of the FFE criterion has been comprehensively validated in previous studies, this research focuses on its application to predict failure under variable loading conditions.

## 1. Introduction

Ensuring the safety and reliability of components is of paramount importance in the aerospace, automotive, and power generation industries. Fatigue failure is among the most common failure modes in mechanical components. Traditional fatigue tests are usually conducted under constant-amplitude sinusoidal loads; however, in practical applications, components face complex loading conditions, including variations in amplitude, frequency, and direction (such as tensile, compressive and torsional loads). This complexity requires a reliable criterion for estimating the lifetime of materials, which has led to the development of fatigue damage accumulation theories.

Fatigue damage accumulation theories are generally classified into linear cumulative theories and nonlinear cumulative theories. In 1945, based on Palmgren’s research and extensive experiments on fatigue damage accumulation, Miner developed the Palmgren–Miner rule [1] (hereafter referred to as the “P-M rule”), which has been widely used in various industries owing to its convenience of application. However, its use in predicting damage under complex loading is challenging because of the assumption that the rate of damage accumulation is constant and that damage due to fatigue from different loads is mutually independent. Various alternative theories for fatigue damage accumulation have emerged to address these limitations. Several reviews [2,3,4] have provided comprehensive overviews of alternative fatigue damage accumulation theories showing that, due to the complexity of random loading, most models fail to fully address the challenges of life prediction in such scenarios. Some studies [5,6] have shown that, under random loading conditions, the fatigue life of specimens is generally shorter than that predicted by constant-amplitude loading experiments. Jensen et al. [7] considered variable-amplitude loading of glass-fiber-reinforced plastics (GFRPs), showing that frequent load amplitude changes will accelerate crack propagation. Additionally, several studies on carbon-fiber-reinforced plastics (hereafter CFRPs) have also indicated that predictions based on the P-M rule are overly optimistic [8,9] Studies have thus introduced the rainflow counting method [10,11] and Dirlik’s empirical probability density function [12] in response. Although these methods reduce errors to an acceptable range, there remains a certain discrepancy with respect to actual conditions.

The entropy-based approach has also garnered attention as an important theory for fatigue damage accumulation [13,14,15,16,17,18,19]. From a thermodynamic perspective, as systems undergo degradation, entropy increases with inelastic deformation, ultimately leading to failure. Therefore, entropy can be used as a tool to quantify the behavior of irreversible degradation, thereby facilitating failure prediction. For many decades, researchers have attempted to use mechanical hysteresis energy as a criterion for fatigue damage, quantitatively linking the fatigue properties of metals to their cyclic stress–strain behavior [20,21]. Building on this, Park and Nelson [22] modified the model to account for the effects of the stress state and incorporated cyclic elastic strain energy density. They applied this enhanced model to multiaxial fatigue life prediction, achieving promising results under certain conditions. Bryant et al. [23] encapsulated the first and second laws of thermodynamics with the Helmholtz free energy, applying this framework to the degradation–entropy generation relationship to relate a desired fatigue measure—stress, strain, cycles, or time to failure—to loads, materials, and environmental conditions such as temperature and heat, through the irreversible entropy produced by the dissipative processes that contribute to material degradation under fatigue. Naderi et al. [24,25] used the entropy generation approach to study the low-cycle fatigue degradation of metals. The hysteresis loop of the stress–strain curve obtained from the experimental results was used to calculate the entropy generation, and the total amount of entropy generation, termed the fracture fatigue entropy (FFE), was introduced as a criterion for fatigue assessment. According to this concept, the FFE is directly related to material type and is independent of load, frequency, and geometry. Liakat et al. [26] proposed a methodology by which the anelastic energy associated with metal fatigue could be determined at a stress level lower than the yield strength of a material, thereby validating the effectiveness of the FFE concept under high-cycle fatigue conditions. Although the FFE-based damage criterion has been developed over a long period and applied in various contexts [27,28,29], including fatigue life prediction under variable frequency and amplitude load conditions [19,25,30,31], most existing research primarily focuses on metallic materials undergoing relatively simple amplitude changes (two- or three-stage). No similar research has been undertaken with relevance to the influence of variable-amplitude loads on polymers and composites [24,32,33,34,35,36,37,38,39].

Previous research has indicated that the lifespan of composite materials largely depends on their resin matrix [40,41]. Consequently, studying the premature failure of the resin matrix under random loading is of significance for determining the lifespan of CFRPs. Considering the FFE criterion’s proven adaptability in life prediction, this study focuses on its application to predict the effects of load history on the premature failure of the resin matrix, rather than focusing on verifying or evaluating the accuracy of the FFE criterion itself. The study begins by introducing the entropy damage criterion, using a generalized Maxwell model previously validated and published by the authors [42], to apply an established approach to fatigue life prediction. Employing this model, a general S–N curve is derived from numerical simulations with constant-amplitude fatigue loading. Based on these numerical results, several variable loading patterns for fatigue failure were determined using the P-M rule. The load sequence in these variable-loading patterns was altered, and their fatigue failure lifespans were numerically predicted. Subsequently, the predicted results are compared with those obtained using the P-M rule. Finally, the specific impact of load history is explored by comparing differences in fatigue life under various load waveforms.

## 2. Numerical Methodologies

### 2.1. Finite Element Simulation

Previous studies have validated that the FFE criterion is independent of the shape or size of a specimen [24]; therefore, the failure criteria for each finite element remain the same, regardless of the shape modeled in the simulation software. This approach differs from real experiments, where the shape and size of a specimen may influence its failure behavior. However, this study does not intend to discuss how models of different shapes undergo damage and failure during fatigue simulations. To simplify the analysis, a single cubic element was modeled for the simulation (as shown in Figure 1).

Abaqus CAE 2020/Abaqus Standard software was used for numerical simulations. The material was modeled as an isotropic thermosetting resin, and a uniaxial tensile load, controlled by stress, was applied along the z-axis (denoted as the 3-direction) to a C3D8 eight-node element. A sinusoidal load with an amplitude ranging from 5 to 60 MPa and a fixed frequency of 2 Hz was applied. Additional constraints were applied in the 1, 2, and 3 directions to prevent displacement or deformation in the non-loading directions. The stress ratio was set to 0 to ensure that the load was entirely tensile with no compression components.

### 2.2. Constitutive Equation Considering Entropy Damage Criterion

Given the constraints in Abaqus for precisely adjusting time, this study indirectly simulates material behavior at different time scales by adjusting the viscosity of dashpot elements. A model consisting of 15 parallel Maxwell elements, each comprising a spring and a dashpot, was employed (Figure 2).

These 15 independent dashpots function across different time scales, allowing the simulation to reflect the broad range of material responses. Stress can be determined by summing all elastic stresses, and stress relaxation occurs individually. According to the viscoelastic constitutive law applied in this study, the total strain ε can be divided into elastic strain $\varepsilon^{e}$ and visco-plastic strain $\varepsilon^{v}$:(1)$$\varepsilon =\varepsilon^{e}+\varepsilon^{v}$$

In Equation (1), the visco-plastic strain $\varepsilon^{v}$ is calculated as
(2)$${\varepsilon^{v}}_{i}=\int_{0}^{t}\frac{E_{i}{\varepsilon^{e}}_{i}}{{g\eta }_{i}}dt$$
where t is time. Then, the visco-elastic constitutive law can be expressed as [14]:(3)$$\sigma \left(t\right)=(1-D)\int_{0}^{t}E^{r}\left(t-t^{\prime }\right)g\frac{d\varepsilon^{e}}{dt^{\prime }}dt^{\prime }$$
where the relaxation modulus **E**^*r*^ is
(4)$$E^{r}\left(t\right)=\sum_{n=1}^{15}E_{ijkl}^{n}e^{-tE^{n}/\eta^{n}}$$
and the nonlinear coefficient g is determined using Equation (5) as a function of von Mises stress *σ_eqv_*, α, m, and *σ*_0_ as follows:(5)$$g=\frac{1}{1+\alpha {\left(\frac{\sigma_{eqv}}{\sigma_{0}}\right)}^{m}}$$

The damage variable *D* can then be calculated using Equation (6) based on entropy generation per unit volume *s* and the final fracture fatigue entropy of the material, denoted as $s^{f}$:(6)$$D=\alpha_{d}\frac{s}{s^{f}}D_{cr}$$

Here, $s^{f}$ (FFE) is a material property, independent of loading conditions such as geometry and loading type [24]. Previous studies generally conclude that material failure occurs when the generated entropy reaches the value of FFE. However, in this simulation, a time acceleration factor $\alpha_{d}$ is introduced, based on prior research [43], to expedite the process [43]. In studies of viscoelastic materials, the time–temperature superposition principle is often used to accelerate the accumulation of damage over a shorter experimental timeframe by incorporating a time acceleration factor, which does not participate in viscoelastic behavior. The value of this factor in real experiments depends on experimental conditions; in this simulation, we determined a suitable time acceleration factor $\alpha_{d}$ based on previous research [43]. The introduction of $\alpha_{d}$ was intended to expedite the simulation. Although the calculated values differ from those in the actual situation, they are used solely for comparison under the same conditions and therefore do not influence the conclusions. Additionally, we set a critical damage threshold, $D_{cr}$, allowing failure to be determined before the entropy generation reaches the original FFE, thereby speeding up the process of analysis.

The calculation of entropy generation in this context directly adopts conclusions from previous studies: first, Helmholtz free energy was used to relate the first law of thermodynamics to the Clausius–Duhem inequality (the second law of thermodynamics). For small strains, the Clausius–Duhem inequality was ultimately simplified into an inequality used to calculate the rate of non-negative generated entropy. The entropy generation rate is determined by three mechanisms: plastic strain energy dissipation, nonrecoverable stored energy in the material, and heat dissipation due to heat conduction. In this simulation, we assume a constant temperature; under low-cycle fatigue, the thermoelastic effect and entropy generated by heat conduction can be neglected. Therefore, the inequality can be further simplified into the following equation [24,30,43,44]:(7)$$s=\int_{0}^{t}\frac{1}{T}\sigma :\dot{\varepsilon^{v}}dt$$
where *T* denotes the absolute temperature, and $\dot{\varepsilon^{v}}$ and ***σ*** come from Equations (2) and (3). Due to the length of the equation derivations and the similarity to content already validated in previous studies, the details are not reiterated here.

Details regarding the numerical model derivation and how the calculation process is integrated into Abaqus as a UMAT are provided in previous studies [42,45]. The parameters used in this study were generated using parameter fitting based on experimental results, and the model used is largely drawn from related prior research (Table 1) [42,43]. It is important to note that although temperature and other variables are often considered in practical scenarios, this study focuses exclusively on the plastic strain characteristics of polyimide materials due to space limitations. Future research should expand the scope of analysis to include additional variables.

## 3. Numerical Results

### 3.1. S–N Curve

Seven stress levels (5, 10, 20, 30, 40, 50 and 60 MPa) were selected for the loading waveform tests to facilitate comparison. These stress values are all lower than the material’s yield stress. To calculate the damage fraction at each stress level based on the linear cumulative damage rule, constant-amplitude load simulations were conducted, and the number of cycles to failure at each stress level was obtained. Using a cyclic load of 20 MPa as an example, Figure 3 illustrates the fatigue ratcheting effect during the first ten loading cycles. The strain was found to gradually increase despite the constant applied stress, indicating a nonlinear damage accumulation process. As the number of cycles increases, the overlapping regions between the loading and unloading paths become denser, indicating that as damage accumulates, the incremental deformation decreases. When the applied stress increases, the effect of the damage becomes more pronounced, i.e., as the number of loading cycles increases, the deformation induced by a single loading increases rapidly, causing the slope of the ratcheting strain curve to decrease and appear more dispersed. The number of failures corresponding to the seven test stresses is summarized in Figure 4 (S–N curve). This curve shows a typical decreasing trend where higher amplitudes result in fewer cycles to failure. As mentioned above, the time acceleration factor $\alpha_{d}$ was introduced to expedite the simulation process, resulting in discrepancies between the simulation outcomes and real values. However, since all simulations were accelerated at the same rate, these differences do not impact the results of the qualitative analysis.

The data used in Figure 4 are shown in Table 2. Using the number of cycles to failure, the fraction of fatigue life expended during one cycle according to the P-M rule can be calculated. Figure 5 shows the damage accumulation before reaching the set damage threshold of 0.25 for each constant-amplitude load, with the horizontal axis representing the fraction of fatigue life expended. The comparison shows that higher-amplitude loads result in greater damage accumulation. Under constant-amplitude loading, the first load causes the most damage, and the damage increments decrease as the cycle numbers increase. At lower stress levels, the damage differences between cycles are smaller, whereas at higher stress levels, the first load accounts for a higher proportion of the total damage. Therefore, the damage increment curve for 5 MPa exhibits a more linear trend than that for 60 MPa.

### 3.2. Variable Loading Patterns

Based on the fraction of fatigue life expended for each stress level in one cycle, the corresponding number of cycles was selected to ensure that the total fraction is as close to 1 as possible. Four distinct loading patterns were constructed with different arrangements, but maintaining a constant total number of cycles for each stress level across all patterns. According to the P-M rule, the order of loading does not affect the number of cycles to failure. Therefore, the resin material is expected to fail at the end of each loading pattern when the fraction of fatigue life expended equals 1.

The structure of Pattern 1 is detailed in Table 3, showing that it follows a simple ascending order: first, 371 cycles of 5 MPa are applied, followed by 189 cycles of 10 MPa, and so on. Pattern 2 (Figure 6a) reduces the number of cycles in Pattern 1 to 1/7 and arranges the stress levels in ascending order to form a block, which is then repeated seven times. In Pattern 3 (Figure 6b), the stress sequence is completely randomized. Pattern 4 (Figure 6c) is intentionally designed in a sequence expected to accelerate failure, such that stress levels are arranged in descending order within a 1/7 block, with 2–3 low-stress levels interspersed among the high-stress levels.

### 3.3. Fatigue Failure under Variable Loading Patterns Compared with the P-M Rule

Figure 7 shows the simulation results after applying four different loading patterns. The results indicate that, at the point of material failure, the fraction of fatigue life expended for the four patterns reached 91.9%, 58.3%, 41.6%, and 40.3%, respectively, compared to the predictions of the P-M rule. Each of these patterns led to faster failure than constant-amplitude loading, suggesting that the conventional S–N diagram may provide overly optimistic and potentially dangerous estimates. When the stress amplitudes were arranged in ascending order without being divided into blocks, the resulting lifespan was closest to that predicted using the P-M rule. However, as the number of stress-amplitude changes increased—whether in the case of loadings divided into seven blocks or in the more frequently varying random-amplitude loading pattern—the fraction of fatigue life expended decreased even further. This finding aligns closely with a previous study [6], which noted that ascending block loading has a minimal impact on lifetime, whereas under random-spectrum loading, the lifespan was reduced to approximately one-third to one-half of that observed under ascending block loading. In Pattern 4, the fraction of fatigue life expended was even smaller than that expended in the random-amplitude loading pattern. Two key conclusions can be drawn from these comparisons: first, variations in stress amplitude accelerate damage accumulation to some extent; second, applying an initial high stress leads to faster failure than applying an initial low stress.

## 4. Discussion

### 4.1. Impact of Historical Load on Damage

To further clarify the previous hypothesis, it is necessary to examine the interactions between different stress levels individually. Taking 60 and 40 MPa as examples, various preloads ranging from 0 to 60 MPa were applied first, followed by a comparison of the damage caused by subsequent 60 and 40 MPa loads. The results (Figure 8) show that, as the preload increases, the damage caused by the subsequent load decreases. Furthermore, when the amplitude of the subsequent stress is lower, the influence of the preload diminishes accordingly. In other words, the impact of the preload on subsequent loading depends on the amplitude of the subsequent stress: the higher the subsequent stress amplitude, the more pronounced the damage-reducing effect of the preload.

Under random variable-amplitude loading, the likelihood of large stress differences between consecutive loads increases. For higher stresses that can cause significant damage, more frequent occurrences of lower preloads lead to greater damage than under constant-amplitude loading. Consequently, to mitigate damage accumulation, it is essential to maintain consistent peak loads and avoid variations in stress amplitude. When the preload is 0, the subsequent load results in the maximum damage. This is a special case, highlighting that the absence of preload results in the most significant effects under subsequent loading.

Based on this finding, it can be surmised why the damage curves for patterns 2 and 4 fluctuate significantly in Figure 7. These fluctuations usually occur at the start of a new loading block, but the reasons for their occurrence are different. In Pattern 2, it is suggested that the fluctuations are not only due to the sudden change in amplitude but also because the influence of the previous high-stress phase persists and is accounted for during the low-stress loading period. In contrast, the fluctuations in Pattern 4 arise due to the long duration of low-stress loading, followed by the onset of high-stress loading in the next block. This loading method significantly increases the damage caused by the application of high stress to the material.

### 4.2. Interrupted Fatigue Simulation

The previous section concluded that the subsequent load causes maximum damage when there is no preload. The absence of a preload can be seen as a pause in loading. To verify the effects of intermittent loading on fatigue life, a new load spectrum—Pattern 5—was created by introducing a 0.5 s pause (equivalent to one cycle) between each load in the simulation, based on Pattern 3. Figure 9 shows a detailed view of a portion of the waveform of Pattern 5. When comparing the damage accumulation data from Pattern 5 with patterns 3 and 4 (Figure 10), it is evident that this combination of random variable-amplitude loading and intermittent pauses further reduces the proportion of fatigue life expended; it reaches only 33.6% of that predicted by the P-M rule.

It is important to note that despite the pause during the loading process, the previously applied stress does not immediately cease to affect the material. In the numerical model, the dashpot component of the model continues to respond, and damage accumulation proceeds at a slower rate. Additionally, in cases like Pattern 4, where a higher stress is intentionally applied first, the damage accumulation rate is accelerated. Following the application of a high stress, it continues to impact the material until failure, although its effect gradually weakens. This implies that, from the start of the fatigue simulation, the earlier and more frequently a material is exposed to higher loads, the faster the damage accumulates, leading to premature failure. Since pause times were introduced in each loading pattern, the horizontal axis is represented as the “fraction of fatigue life expended” rather than “time to failure” when comparing material life under different patterns. This choice is significant because, when pauses are included, the lowest fraction does not necessarily correspond to the shortest time to failure. However, this representation makes it easier to compare the amount of external work required to cause material failure under different patterns. As outlined above, Pattern 5 includes a pause of only 0.5 s, suggesting that it is still possible to further reduce the number of stress cycles to failure. Simulation results show that if the pause time is slightly extended, the fraction of fatigue life expended will continue to decrease. Since pauses in loading allow the material some time to rest, the accumulation of fatigue damage slows down as the pause time increases. Once the pause time reaches a certain length, the time to fatigue failure will cease to decrease. This minimum failure time depends on the response characteristics of the dashpots, meaning that it is closely related to their material properties. The introduction of pause time provides a novel approach for optimizing fatigue life prediction methods for polymer matrix and CFRP materials in future research.

## 5. Conclusions

In this study, several variable-amplitude loading patterns were designed based on the S–N curve, and fatigue failure simulations were conducted. The results obtained were compared to the fatigue life predicted by the Palmgren–Miner (P-M) rule. The main findings of this study are as follows:Frequent changes in stress amplitude accelerate damage accumulation to some extent; applying an initial high stress leads to more rapid failure compared to applying low stress first.As the preload increases, the damage caused by subsequent loads decreases. The effect of different preloads on subsequent loads depends on the amplitude of the latter, with higher amplitudes showing a more pronounced damage-reducing effect. When there is no preload, the load causes maximum damage.Intermittent pauses further reduce the proportion of fatigue life expended.Pauses also delay fatigue, such there is a maximum pause time leading to the fastest failure, which is closely related to material properties.

Under the variable-amplitude loading patterns considered in this study, the proportion of fatigue life expended at failure ranged from 33.6% to 91.9%. The predicted fatigue life varies significantly due to changes in loading sequence, but also shows cases where it falls far below the values predicted by the P-M rule. Consequently, this study offers a more practical and reliable approach for predicting fatigue life under complex loading conditions. Future research should explore a broader range of materials and loading scenarios to further optimize and refine the accuracy of the proposed prediction model.

## Figures and Tables

**Figure 1 polymers-16-02857-f001:**
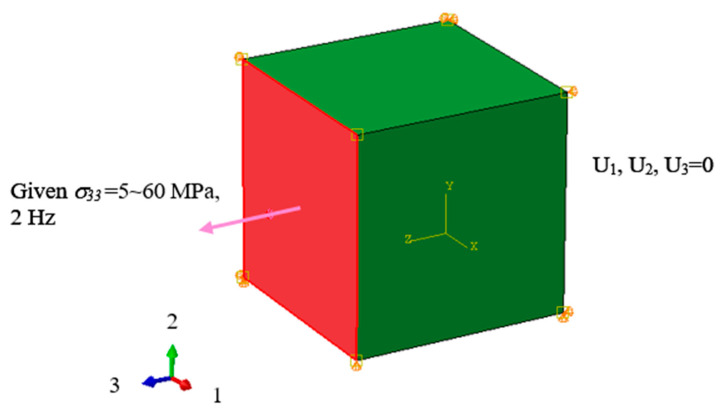
Boundary conditions applied to the simulated element.

**Figure 2 polymers-16-02857-f002:**
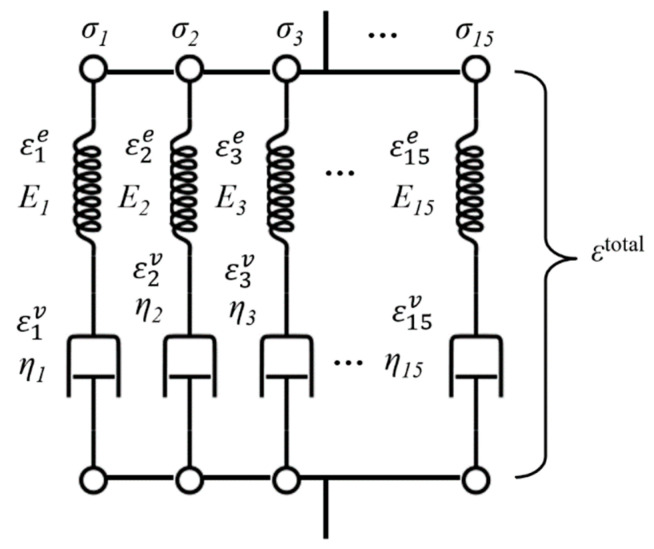
Numerical model for the constitutive equation.

**Figure 3 polymers-16-02857-f003:**
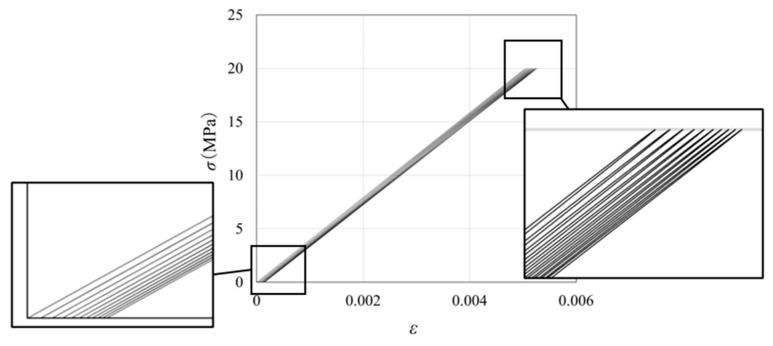
Stress–strain hysteresis loop with stress amplitude of 20 MPa for 10 cycles.

**Figure 4 polymers-16-02857-f004:**
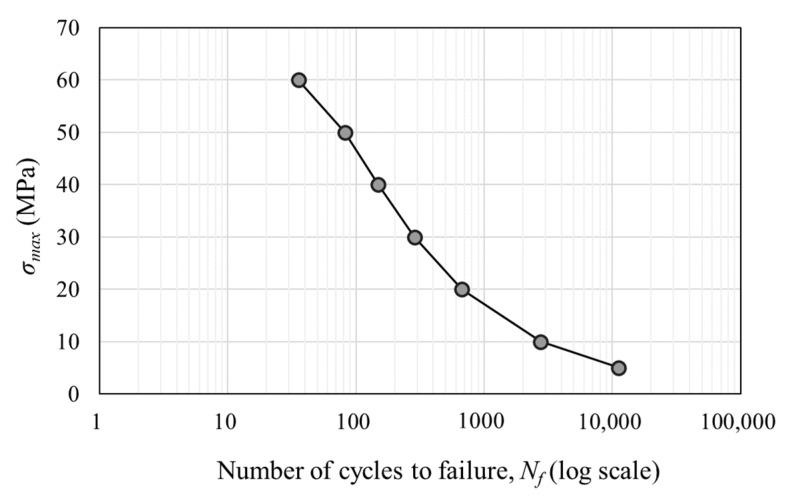
Relationship between load level and number of cycles to fatigue failure.

**Figure 5 polymers-16-02857-f005:**
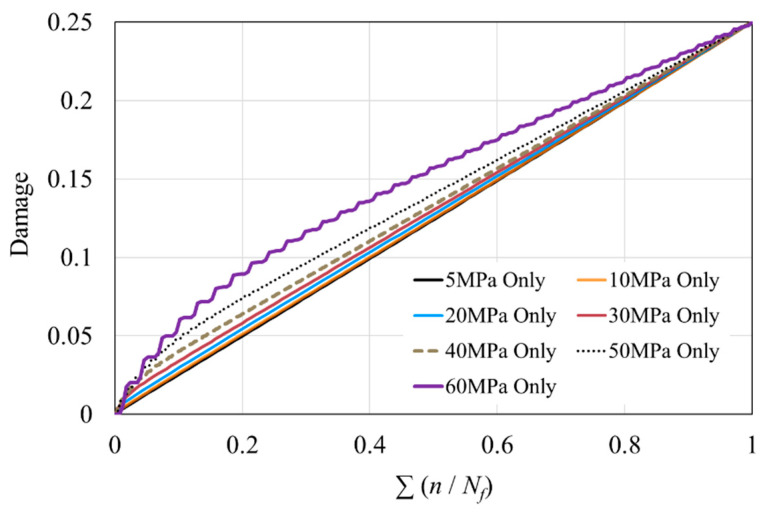
Damage accumulation based on P-M rule.

**Figure 6 polymers-16-02857-f006:**
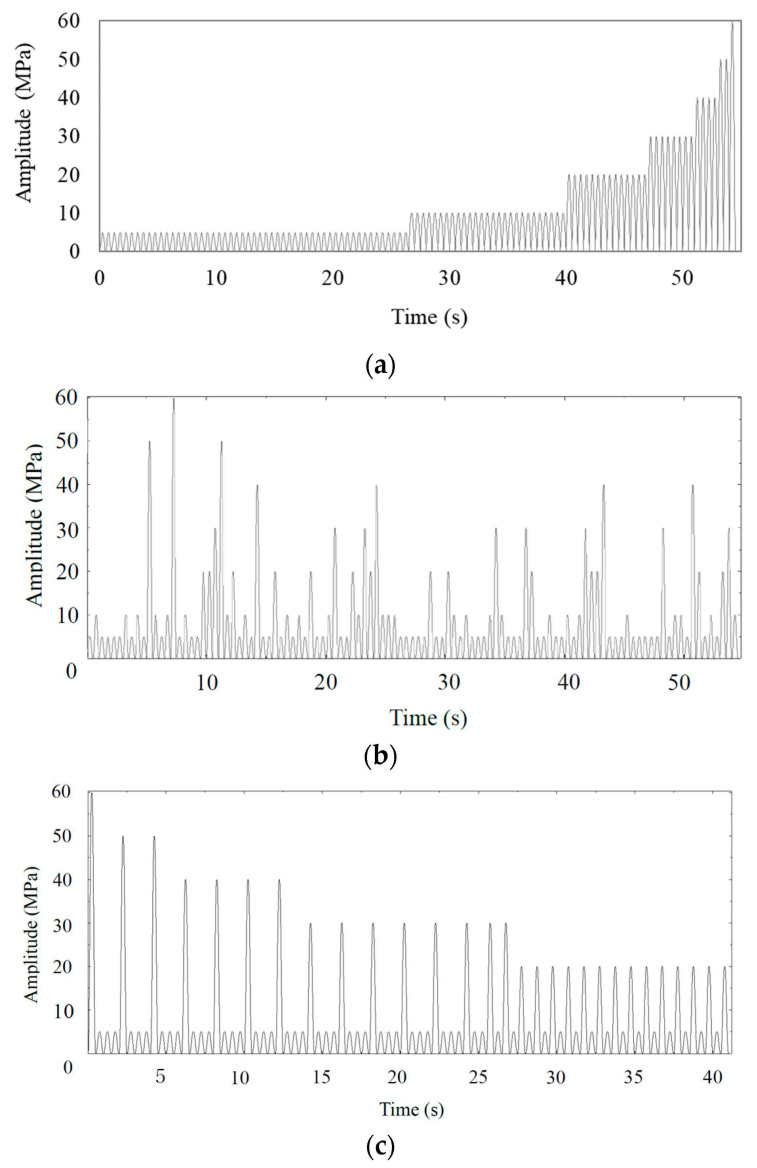
Waveforms when the fraction of fatigue life expended = 100% (a block). (**a**) Pattern 2, characterized by a loading sequence, in order, from low stress of 5, 10, 20, 30, 40, 50, and 60 MPa for 53, 27, 14, 8, 4, 2, and 1 cycles; (**b**) Pattern 3, characterized by a random sequence (a part of which is shown); (**c**) Pattern 4, consisting of the same amplitudes as Pattern 2, but with accelerated failure sequences (except 10 MPa).

**Figure 7 polymers-16-02857-f007:**
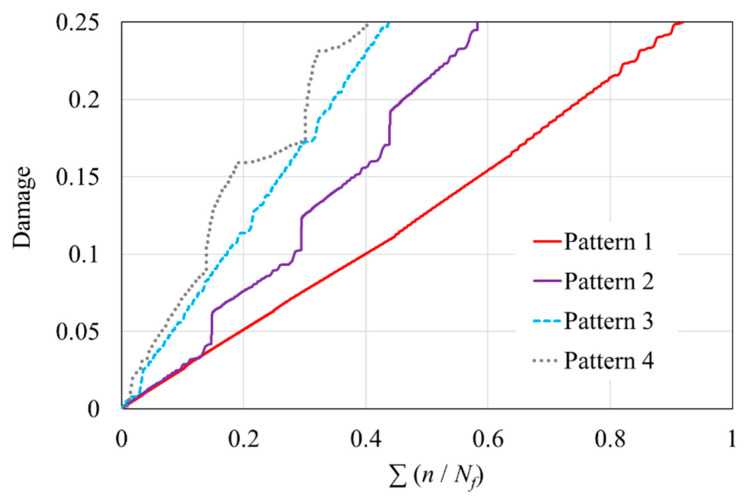
Fraction of fatigue life expended under four loading patterns.

**Figure 8 polymers-16-02857-f008:**
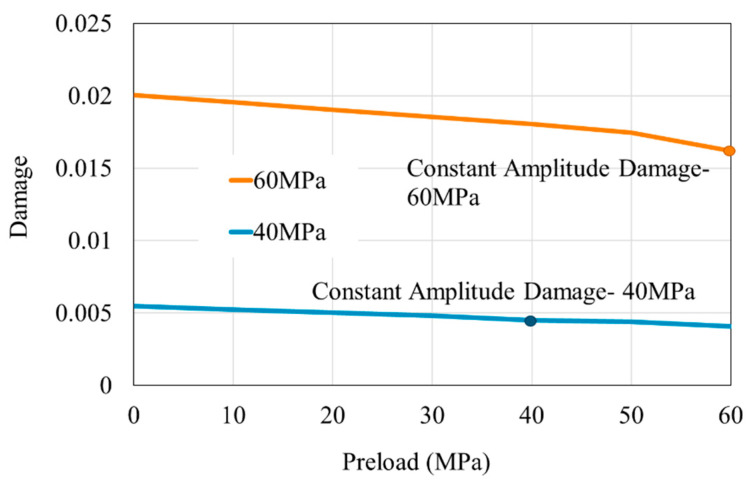
Damage increments due to subsequent loads following different preloads.

**Figure 9 polymers-16-02857-f009:**
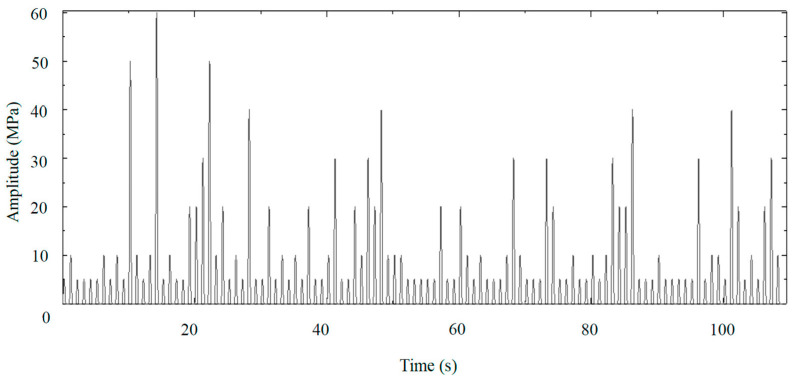
Waveforms with pauses when the fraction of fatigue life expended reaches 100% (Pattern 5).

**Figure 10 polymers-16-02857-f010:**
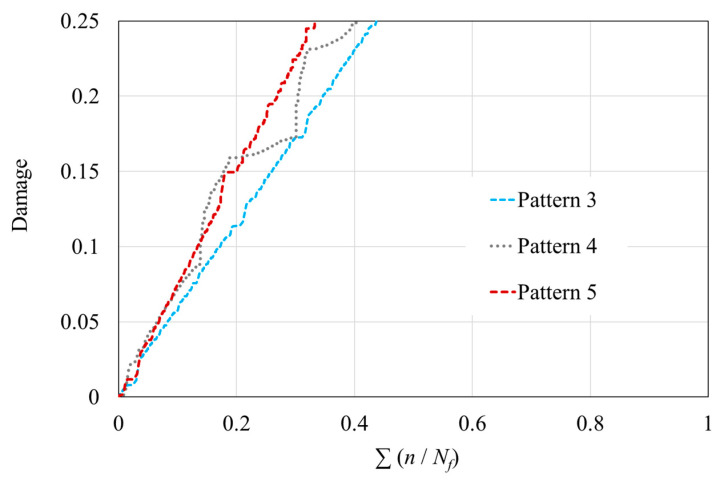
Fraction of fatigue life expended under Pattern 5 compared to patterns 3 and 4.

**Table 1 polymers-16-02857-t001:** Properties of the materials used in this study [42,43].

n	$E_{n}(\mathrm{MPa})$	$\eta_{n}(\mathrm{MPa}\cdot s)$	Elasticity	
1	284	4.5 × 10^2^	$E_{0}(\mathrm{MPa})$ *	4260
2	284	3.3 × 10^3^	*ν* *	0.3
3	284	1.2 × 10^5^	**Nonlinearity**	
4	284	1.9 × 10^6^	$\sigma_{0}(\mathrm{MPa})$	70
5	284	1.8 × 10^7^	*α*	2
6	284	1.4 × 10^8^	*m*	7
7	284	8.5 × 10^8^	**Entropy-relating**	
8	284	5.0 × 10^9^	*s^f^*	0.1 J/m^3^ · K
9	284	3.0 × 10^10^	*D_cr_*	0.25
10	284	1.9 × 10^11^	*T*	303 K
11	284	1.4 × 10^16^		
12	284	1.3 × 10^19^		
13	284	2.1 × 10^22^		
14	284	1.3 × 10^26^	**Damage acceleration parameter**	
15	284	2.5 × 10^29^	$\alpha_{d}$	4

* $E_{0}$, *ν*: initial Young’s modulus, Poisson’s ratio.

**Table 2 polymers-16-02857-t002:** Number of cycles to failure and the fraction of fatigue life expended according to the P-M rule per cycle under each stress condition.

*σ*_max_ (MPa)	Number of Cycles to Failure (Cycle)	1/*N_f_*
5	11,169	0.01%
10	2760	0.04%
20	671	0.15%
30	286	0.35%
40	150	0.67%
50	82	1.22%
60	36	2.81%

**Table 3 polymers-16-02857-t003:** Loading sequence from low to high based on the P-M rule (Pattern 1).

*σ*_max_ (MPa)	Number of Cycles (Cycle)
5	371
10	189
20	98
30	56
40	28
50	14
60	7

## Data Availability

The raw data supporting the conclusions of this article will be made available by the authors on request.

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
