# Peer review of "Numerical Simulation for Durability of a Viscoelastic Polymer Material Subjected to Variable Loadings Fatigue Based on Entropy Damage Criterion"

_polymers, 2024, doi:10.3390/polym16202857_

Round 1
Reviewer 1 Report
Comments and Suggestions for Authors
This research aims to explore the impact of load history on the premature failure of the viscoelastic polymer matrix in carbon-fiber-reinforced plastics (CFRPs) using a method based on the framework of the fracture fatigue entropy (FFE) concept. The results show that both frequent amplitude changes and even brief pauses in loading accelerate damage accumulation and lead to premature failure of the polymer matrix. In summary, the research provides a certain reference for the development of more durable and efficient composite materials.
1. There are significant issues with references, as the citations for references 53 and beyond are missing.
2. The boundary conditions in Figure 1 are not clear, and the direction of the coordinates is not parallel to the direction of the square. Please redraw the figure.
3. The title of Section 2.1 seems unsuitable. It is suggested to revise it to 'Finite Element Simulation.'
4. In line 152, the citation of reference is suggested to consolidate multiple references within a single set of parentheses. Please also revise other citations throughout the text accordingly.
5. For lines 195-199, please provide relevant references and further elaborate on the reasons why the introduction of the time-accelerating factor does not affect the simulation results.
6. Table 2, please use “Number of cycles to failure (
)” as the header of the second column.
7. Figure 4 should include labels for the logarithmic axes.
8. The Figure 4(a) seems different from Figure 4(b) and (c), please redraw the figure.
9. For lines 251-254, please explain in more detail why the volatility of mode 4 is much greater than that of the other modes
10. Let the name of Figure 9 like that of Figure 6.
11. Please refine the abstract and conclusion sections further.

1. The title of Section 2.1 seems unsuitable. It is suggested to revise it to 'Finite Element Simulation.'
2. Please refine the abstract and conclusion sections further.
Author Response
Thank you very much for taking the time to review this manuscript. Please refer to the attached files for detailed responses and the revised manuscript.

Reviewer 2 Report
Comments and Suggestions for Authors
In the research article titled “Numerical Simulation for Durability of a Viscoelastic Polymer Material Subjected to Variable Loadings Fatigue Based on Entropy Damage Criterion” Li et al., have comprehensive study of a user-defined subroutine (UMAT) to observe the FFE damage criterion within the ABAQUS software. There are few aspects in the study which should be addressed before being published, which are as follow;
1. The abstract section is providing the general discussion about the measurements authors have made in the study, there should be outcomes (values) descriptions to defend the results and make the abstract attractive. I suggest abstract should be re-written with a sentence describing the applications of this study in the end of abstract.
2. In the introduction section, authors have referenced many articles and described what previous authors have done about this work, but introduction is weak in a sense like
I. Park and Nelson[25] modified the model to account………..achieving promising results under certain conditions
II. Doelling et al. [26] proposed that the ire…………versible entropy variable for quantitatively describing degradation
III. Naderi et al.[28,29] used the entropy generation approach…….is used to calculate the entropy generation
In all these studies authors have not described their outcomes (values), so that the comparison can be made. Authors should make the introduction section attractive rather than just formal discussion.
3. What are the significance if Eq. 1, Eq 2, Eq.3and Eq. 7? Authors have not described anything about these equations in their explanation. We can understand their importance but for the new beginners their purpose should be properly mentioned in the literature and also their references should also be there.
4. Explanation of Figure 4 is confusing, please elaborate it further.
5. There exist several grammatical, typo and conceptual errors in the manuscript, which must be revised by the Native English speaker of same field like
· Line 194, full stop is missing
· Line 200, The specific numbers are shown in Table 2. (which specific numbers?)
· Line 218 is as close to 1 as possible
· Etc.
6. Figure 6 should be combined, their a, b c sections should be taken inside the Figure, alongwith a complete Figure 6 caption.
7. There should be a comparison table before the conclusion, which should mention the comparison of this article’s results with the previous literature.
Comments on the Quality of English Language
English should be revised.
Author Response

(The authors gave the same response as above.)
